# Evaluations of Quality by Design (QbD) Elements Impact for Developing Niosomes as a Promising Topical Drug Delivery Platform

**DOI:** 10.3390/pharmaceutics12030246

**Published:** 2020-03-09

**Authors:** Parinbhai Shah, Benjamin Goodyear, Anika Haq, Vinam Puri, Bozena Michniak-Kohn

**Affiliations:** 1Department of Pharmaceutics, Ernest Mario School of Pharmacy, Rutgers, The State University of New Jersey, Piscataway, NJ 08855, USA; shah.parin@rutgers.edu (P.S.); benjamin.goodyear@rutgers.edu (B.G.); anika.alam@rutgers.edu (A.H.); vinam.puri@rutgers.edu (V.P.); 2Center for Dermal Research, Life Science Building, Rutgers, The State University of New Jersey, Piscataway, NJ 08854, USA

**Keywords:** topical drug delivery, niosome, quality by design (QbD),, Critical Material Attributes and Critical Process Parameters (CMA and CPP), desoximetasone, controlled drug delivery, allergic reactions, eczema, psoriasis

## Abstract

Topical corticosteroids are used to treat a variety of skin conditions such as allergic reactions, eczema, and psoriasis. Niosomes are a novel surfactant-based delivery system that may be used to deliver desoximetasone via topical product application in order to mitigate common side effects associated with traditional oral delivery routes. The aim of this research was to identify the critical material attributes (CMAs) and critical process parameters (CPPs) that impact key characteristics of drug-loaded niosomes using a systematic quality by design (QbD) approach. An organic phase injection method was developed and used to manufacture the niosomes. The CMAs were identified to be drug amount, concentrations of surfactant and cholesterol, and types of lipids. The CPPs were phase volumes, temperature, mixing parameters, and addition rate based on previous research. The quality attributes measured were entrapment efficiency, particle size distribution, PDI, and zeta potential. These were used to determine the quality target product profile (QTPP) of niosomes. The experimental data indicate that the critical impacting variables for niosomes are: surfactant and cholesterol concentrations, mixing parameters, and organic-phase addition rate. Based on the experimental results of this study methanol:diethyl ether (75:25) as the organic system, drug:surfactant:cholesterol in 1:2:1 concentration, stearic acid as the charge-inducing material, 20 mL external phase and 10 mL internal phase volume, 65 °C external phase temperature, 60 min mixing time, 650 RPM mixing speed and 1 mL/ml addition rate is the ideal combination to achieve desirable desoximetasone niosomes with optimum entrapment efficiency and particle size for topical application.

## 1. Introduction

Drug delivery systems play an important role in dermatological product development phases for producing effective treatments for patients [1]. Controlled drug delivery systems may be used to achieve modified drug release kinetics with targeted site-specific delivery which increases therapeutic efficacy, patient compliance, and improves adherence to patient protocols. The concept of site-specific drug delivery was introduced to the pharmaceutical industry by Paul Elrich in 1909 [2]. One of his well-known works reported a ‘magic bullet’ to deliver a drug to a desired site of action with no significant interactions to non-target organs or local tissues. The core purpose of a site-specific drug delivery system is not only to increase the selectivity but also to reduce the systemic toxicity of the drug. In order to direct a therapeutic active pharmaceutical ingredient to a particular site of action in a controlled manner, researchers must also consider all the potential risks of systemic exposure and toxicity during the initial formulation process.

An ideal drug delivery system should carry the therapeutic agent to the site of action and adequately release it over a predetermined amount of time [3]. Different carriers have been successfully developed and used especially for targeted drug delivery that include: synthetic polymers, immunoglobulins, serum proteins, liposomes, niosomes, microspheres, and erythrocytes, etc. Colloidal particulate carriers such as liposomes and niosomes as drug delivery systems have distinct advantages over conventional dosage forms [4]. They can act as drug reservoirs from which the drug can be released at the targeted site and the drug release can be modified by changing their composition [5]. 

Over the past twenty years, researchers have observed some key benefits when deploying liposome and niosome technologies in topical drug delivery applications for both localized and systemic absorption routes [6]. Currently, niosome technologies are broadly studied as an alternative to solve the drawbacks of liposomes as carrier systems. 

The high formulation cost and limited shelf life associated with liposome formulations are the main factors which eventually lead to the development of vesicular systems that may overcome major drawbacks of liposomes. Niosomes or non-ionic surfactant vesicles are alternatives which share some similar properties and components when compared to liposomes. 

Niosomes are highly organized microscopic drug containing-vesicles and measure between 10 and 1000 nm in diameter. They can be unilamellar or multi-lamellar vesicles, wherein an aqueous solution is enclosed by a bilayer structure consisting of non-ionic surfactants, such as Span 20, 40, 60 and 80, Tween 20, 60, 61 and 80, Brij^TM^ with or without cholesterol [7,8,9,10]. These structural properties of niosomes allow the entrapment of drugs with varying lipophilicities. The core can contain both hydrophilic drugs, which are entrapped in the hydrophilic core, and lipophilic drugs, which are distributed completely around the bilayer. Drugs with intermediate log *p* values are appropriately distributed between the hydrophilic and hydrophobic phases (both in the bilayer and in the hydrophilic core). 

The addition of cholesterol with the non-ionic surfactant provides the rigidity and orientational order to the niosomal bilayer [11]. In addition, the fluidity of the niosomal bilayers is dependent on the hydrophilic head group, alkyl chain length and cholesterol content [11]. Niosomes have been successfully used in drug targeting to various organs, such as the skin, brain, liver, lung and ocular systems [12,13,14]. Additionally, in the past two decades, niosomes have been investigated as potential drug delivery systems for anticancer, anti-inflammatory, antifungal and topical/transdermal drug delivery [15,16,17,18]. 

Recently, niosomes have attracted a great deal of attention in the field of topical/transdermal drug delivery due to their many advantages, such as biodegradability, non-toxicity, non-immunogenicity, amphiphilic nature, possibility to modulate drug bioavailability [19], enhancement of drug penetration, provision of a solubilizing matrix and a local depot in deeper layers of skin for sustained drug release. Additionally, topical application of niosomes can increase the residence time of drugs in the stratum corneum and epidermis, thereby increasing the skin deposition while reducing the systemic absorption, since they act as a rate-limiting membrane [17,20]. A study conducted by Manca et al. showed that all experimental niosomes were internalized in human keratinocytes in vitro with a different mechanism of cellular uptake, which seems to be affected by the vesicle composition [21]. When applied topically, niosomes show a desired interaction with human skin by improving the stratum corneum layer characteristics due to reduction in transepidermal water loss and increase in smoothness via replenishing lost skin lipids [22,23]. Moreover, the adsorption and fusion of niosomes on the skin surface can lead to a higher thermodynamic activity gradient of the drug (the driving force for permeation) at the interface and therefore can influence the permeation of lipophilic drugs [24]. A variety of drugs, including estradiol, tretinoin, benzoyl peroxide, diclofenac sodium, ketoconazole, gliclazide, etc., have been successfully encapsulated in niosomes for dermal delivery [25,26,27,28].

Topical corticosteroids are used to treat a variety of dermatological disorders such as atopic dermatitis, eczema, and psoriasis. Encapsulation of corticosteroids in an appropriate carrier system can improve therapeutic efficacy and drug targeting by reducing adverse effects and by increasing patient compliance. Desoximetasone is a drug used to treat various localized skin conditions including eczema and psoriasis by reducing inflammation and relieve itching. This drug has a molecular weight of 376.468 g/mol, logP of 2.35 and aqueous solubility of 0.031 mg/mL. Due to these properties desoximetasone is not an ideal candidate for conventional dermal drug delivery applications due to the poor drug solubility in water. A niosomal-based drug delivery system may be a potential way to approach improving drug solubility and therapeutic drug permeation to target skin tissue. Adelaide et al. presented a comparative study between tacrolimus 0.1% ointment versus desoximetasone 0.25% and tacrolimus 0.1% ointment combination product for the treatment of the atopic dermatitis where they used desoximetasone for the synergistic effect [29]. Various desoximetasone containing marketed products are available in ointment, cream, gel dosage form for the treatment of atopic dermatitis [30]. To the best of our knowledge, there are no published data that demonstrate the preparation of desoximetasone-loaded niosomes. 

During niosome preparation, various formulation and processing variables can influence the performance of the final product. Thus, the study of these variables in niosome preparation will be a valuable addition to the scientific information provided about these carriers. The quality by design (QbD) approach encompasses designing and developing a product in which manufacturing processes meets predefined product criteria [31]. It is a systematic approach to recommend that quality should be built into the process and product during development, going beyond the traditional quality by testing (QbT) approach, where the quality is mainly tested in the final product [32]. It can be used to study the effect of several factors influencing responses by varying them simultaneously by carrying out a limited number of experiments. Thus, by using this approach, the costs and time associated to a drug’s development and manufacturing process can be significantly reduced [32]. Moreover, it is useful in obtaining the “best possible” formulation composition and provides holistic understanding of the process and product behaviors [33]. The important part of this approach is to understand how critical material attributes and process parameters affect the product quality and subsequent optimization parameters with respect to the final specifications [34]. This advanced approach is being widely promoted by the Food and Drug Administration and the International Conference on Harmonization (ICH) [35]. Additionally, QbD elements are now regulatory requirements of the submissions [36]. 

The aim of this research was to prepare desoximetasone-loaded niosomes using an organic phase injection technique to identify the critical material attributes (CMAs) and critical process parameters (CPPs) that impact the key characteristics of drug-loaded niosomes using a systematic QbD approach and to characterize the formulations for the drug amount, entrapment efficiency, particle size, PDI and zeta potential. The experimental data in this investigation suggest that the critical impacting parameters for niosomes are surfactant and cholesterol concentrations, mixing parameters, and rate of addition [37,38].

## 2. Materials and Methods

### 2.1. Materials

Desoximetasone was gifted by Flavine, New Jersey, USA. Diethyl ether and stearyl amine were procured from Sigma-Aldrich, Saint Louis, MO, USA. Ethanol was procured from Decon Labs, Inc., King of Prussia, PA, USA. Methanol, acetone and acetonitrile were purchased from BDH VWR Analytical, Radnor, PA, USA. Span 60 and cholesterol were gifted from Croda Inc., Mill Hall, PA, USA. Stearic acid was obtained from BASF corporation, Edison, NJ, USA. HPLC water and chloroform were purchased from Sigma-Aldrich, Saint Louis, MO, USA. Glacial acetic acid was procured from Fisher Scientific, Fair Lawn, NJ, USA.

### 2.2. Niosome Vesicle Preparation

The drug was dissolved into an organic phase, then it was mixed until completely dissolved. Next, sorbitan monostearate (Span 60), cholesterol, and lipid were added into the solution and mixed using a magnetic spin bar in a 20 mL glass scintillation vial. In a separate 50 mL glass beaker, purified water was heated at various temperatures using a hot plate with magnetic stirring. The temperature of the water phase was selected based on the design requirement. The organic phase was filled into a 10 mL syringe with a 26 G needle. The organic phase mixture was injected into the preheated aqueous phase using predetermined parameters based on the experimental design. Mixing was carried out based on the values identified from the design of experiment (DoE). In the last step of the process, the batch was cooled down to RT and the formulation was stored in a suitable glass storage container. 

### 2.3. HPLC Methods

The mobile phase was prepared by mixing methanol, HPLC grade water and glacial acetic acid (65:35:1). The diluent was prepared by mixing methanol and acetonitrile (50:50). The desoximetasone was measured using a Discovery C18 column with 5 µm particle size, L × I.D. 150 mm × 4.6 mm and UV absorbance set to 254 nm. The injection volume was 10 µL and the flow rate was 1.0 mL/min. The sample run time was 10 min at room temperature and retention time for the desoximetasone elution peak was at approximately 5 min. The linearity measurements were performed as per the ICH guideline with 6 serial dilution range from 0.0025 to 0.08 mg/mL with a R^2^ value of 0.999.

### 2.4. Optimization of Drug Loaded Niosomes Using a Systematic Experimental Design

Known elements were identified through a literature search and published manuscripts/patents. Based on past publications describing vesicular drug delivery systems and formulation variables, material attributes (organic phase, drug concentration, surfactant concentration, cholesterol concentration and types of lipid) and process parameters (internal and external phase volume, external phase temperature, mixing time, mixing speed and addition rate) were selected for further study. Variables used in the formulation design are described in Table 1 and details on the design of experiments are provided in Table 2.

The design of experiments provided in Table 2 was generated using a QbD approach. The two categories shown are the niosome formulation CMAs and the CPPs considered in this study. Table 2 also highlights the control elements and outlines the 29 formulations tested for the 11 critical elements studied using previously described manufacturing methods. The process parameters along with manufacturing processing controls for the niosomal dispersion were the following: 20 mg drug content, diethyl ether:methanol (7.5:2.5) organic phase solvent, 40 mg surfactant, 20 mg cholesterol and 5 mg stearic acid. Standard control parameters were well maintained using an external phase temperature of 65 °C, 20 mL sample volume, 10 mL internal phase volume, 650 rpm mixing speed, 60 min total mixing time, and a constant organic phase-injection addition rate of 1.0 mL/min.

### 2.5. Niosome Vesicle Characterization

#### 2.5.1. Organoleptic Properties

Niosomal dispersions were characterized for visual appearance, color, and odor to confirm the presence of any residual solvent standard quality check.

#### 2.5.2. Drug Content (Assay Determination)

The desoximetasone niosomal dispersion was carefully collected and placed into an intermediate solvent containing a chloroform:methanol (40:60) mixture and then mixed using a vortexing mixer until it was completely dissolved at room temperature. Upon mixing, niosomal dispersion samples were further diluted with equal ratios of diluent. Drug content quantification for desoximetasone was done using HPLC (Agilent 1100–Chemstation software) coupled with UV analysis at a wavelength of λ_max_ 254 nm.

#### 2.5.3. Drug Entrapment Efficiency of Niosome Vesicles

The desoximetasone free drug was determined from entrapped drug by ultracentrifugation at 14,000 rpm for 30 min using an ultracentrifuge (Branson Ultrasonics Corporation, CT, USA) at room temperature (centrifugation speed was validated using 3750 rpm and 14,000 rpm and centrifugation time was validated using 30 min and 60 min). Supernatant containing the free drug was carefully collected without disturbing the sediment of the sample. The supernatant was dissolved into chloroform:methanol (40:60) mixture using a vortex mixer. After mixing, the sample was further diluted with an equal amount of the diluent. Drug quantification was determined using the pre-determined HPLC method. Drug % entrapment efficiency was calculated in triplicate by using the following formula [25].
$$\%\;\mathrm{Entrapment}\;\mathrm{Efficiency}=\frac{Total\;amount\;of\;Drug-Free\;amount\;of\;Drug\;}{Total\;amount\;of\;Drug}\times 100$$


#### 2.5.4. Niosomes Vesicle Size and Zeta Potential

The mean vesicle size and its distribution were estimated at room temperature using a Delsa Nano S Particle Sizer (Beckman Coulter, CA, USA) in triplicate based on light scattering spectroscopy principles. The zeta potential of the niosomal suspensions was measured in triplicate using a Malvern Particle Sizer 2000 (Malvern Technologies, Worcestershire, UK). 

## 3. Results and Discussion

Niosome formulations may be directly impacted by process parameters, surfactant chemistries, presence of membrane additives, and physiochemical drug specific attributes. In order to guarantee drug entrapment into the vesicle matrix, commonly used surfactants with alkyl-chain lengths with C12-C18 are best suited to ensure stable and robust niosomes, as demonstrated in previously published work [39]. Span 60 (sorbitan monostearate) is used solely as surfactant for the preparation of niosomal formulation due to its longer saturated alkyl chain, which shows higher entrapment efficiency compared to other members of the Span family. Additionally, the higher phase transition temperature and lower HLB (4.7) of Span 60 are key elements for its higher entrapment efficiency capacity [40]. 

Niosomes containing desoximetasone were prepared using the non-ionic surfactant Span 60, cholesterol and lipid by considering method of preparation, drug concentration, surfactant concentration, cholesterol concentration, selection of lipid as CMAs and external phase temperature, external phase volume, internal phase volume, mixing time, mixing speed and addition rate as CPPs. The preparation of niosomes was evaluated for entrapment efficiency, particle size, polydispersity index and zeta potential. 

### 3.1. Organoleptic Properties

Niosomal formulations are often described as being milky white in color, odorless dispersions (see page 13) with a fluid-like consistency. Niosomal dispersions were obtained in all the formulations except with formulation DND-1. The drug is not soluble in diethyl ether [41], and, due to this fact, a niosomal dispersion was not obtained. 

### 3.2. Optimization of Niosomes by Design of Experiments (DoE)

Niosome formulations were prepared by deploying a design of experiments approach. The effect of various formulation variables included: drug concentration, solvent system, surfactant concentration, cholesterol concentration, type of lipid, external phase temperature, external phase volume, internal phase volume, total mixing time, mixing speed and addition rate. 

The combination of independent elemental variables resulted in an observable change in entrapment efficiency, niosome size, polydispersity index and zeta potential.

### 3.3. Optimization of Critical Material Attributes

#### 3.3.1. Effect of Organic Phase System

In the of various solvent systems effect on niosomes, it was observed that niosomes did not form with diethyl ether as a solvent system. This was due to the fact that desoximetasone was insoluble in diethyl ether. Drug entrapment efficiency was comparatively lower when ethanol or acetone were used as a solvent system. When the solvent system was changed from a diethyl ether:ethanol combination to diethyl ether:methanol, it resulted in a higher drug entrapment efficiency. These data are provided in Table 3. The impact of the organic phase solvent system on entrapment efficiency and particle sizes is illustrated in Figure 1.

#### 3.3.2. Effect of Drug, Surfactant and Cholesterol Concentrations

The formulations containing 2:4:2, 4:4:2, 2:4:3 ratios of drug:surfactant:cholesterol prepared using the diethyl ether: methanol solvent system showed higher entrapment efficiency in comparison to the formulations containing 3:4:2, 2:2:2, 2:3:2, 2:5:2, 2:6:2, 2:7:2, 2;4:4 and 2:4:5 ratios of drug:surfactant:cholesterol. It was observed that the relative amount of span 60 and cholesterol played a key role in determining the drug loading into the niosomal matrix. It was clearly observed that with an increase in surfactant concentrations, entrapment efficiency increased, up to and including a 2:4:2 ratio of drug:surfactant:cholesterol. Beyond this point, an increase in cholesterol content showed a decrease in entrapment efficiency and an increase in particle size. This observed behavior may have been due to two factors: (i) higher cholesterol increases the bilayer hydrophobicity and stability and decreases bilayer permeability, which may lead to efficient hydrophobic drug entrapment in bilayers as the vesicle formed [42]; (ii) higher cholesterol content makes the niosome structure more rigid and increase the particle size while competing with the drug for packing space within the bilayer. A similar trend was also observed and reported in the various literatures [40,43,44]. Impact of surfactant and cholesterol concentrations on entrapment efficiency and particle size are provided in Figure 2 and Figure 3, respectively.

#### 3.3.3. Influence of Electrostatic Charge

Niosomes were formulated with a wide variety of charged surfactant materials. Control niosomes were also prepared without charged materials to evaluate their impact on the formulation. The inclusion of a charge-inducing agent in the lipid layer prevents the aggregation and fusion of vesicles, and maintains their integrity and uniformity [44]. The optimum charge-inducing agent was selected based on the entrapment efficiency and particle size of the niosomes. Stearylamine, which has a positive charge, produced niosome particle sizes about 56,000 nm that were not acceptable, and the PDI could not even be measured. This behavior was due to the fact that stearylamine was not able to create enough repulsion force between the niosome particles. The experiments revealed that 5 mg stearic acid was sufficient to produce uniformly dispersed niosomes with higher drug entrapment efficiency, selective particle sizes and homogenous distribution, as shown in Table 3.

### 3.4. Optimization of Critical Process Parameters

#### 3.4.1. Effect of Volume and Temperature

We examined the effect of external phase temperature and volume on niosomal formulations containing drug:surfactant:cholesterol with the ratio 2:4:2 with the diethyl ether:methanol solvent system and stearic acid as the charge-inducing agent. Various external phase temperatures (55 °C, 65 °C and 75 °C) and external phase volumes (10 mL, 20 mL and 30 mL) were tried. The results demonstrate that external phase temperature had no discriminative impact on particle size and drug entrapment efficiency of the niosomal formulations. The entrapment efficiency observed was comparatively lower in formulations processed with an external phase temperature of 75 °C. This behavior may have been caused by higher temperatures interacting with the drug molecule stability. Entrapment efficiency and particle size were found to be the same when comparing formulations made with external phase volumes of 10 and 20 mL. It was observed that were was a decreasing trend in entrapment efficiency and an increasing trend in particle sizes with the increase in external phase volume above 20 mL. This behavior can be explained by the possibility that increased hydration volume may have increased drug leakage from the niosomes, leading to a decrease in entrapment efficiency [45]. 

#### 3.4.2. Effect of Mixing Time (minutes) and Speed (rpm)

The niosomal formulations containing a drug:surfactant:cholesterol ratio of 2:4:2 with the diethyl ether:methanol solvent system and stearic acid as a charge-inducing agent were subjected to various mixing times (30, 45 and 60 min) and mixing speeds (450, 550 and 650 rpm). The data suggest that increasing mixing speeds from 450 to 650 rpm and mixing times from 30 to 60 min resulted in higher entrapment efficiency and lower particle sizes. This behavior may be explained by the fact that higher mixing speed leads to faster evaporation of the solvent system, which in turn resulted in uniform niosomes with higher entrapment efficiencies. The results also demonstrate that lower mixing times were not sufficient enough to complete the formation of niosomes. This may result in more drug staying in the free form. Comparisons of mixing time and mixing speeds and their effects on entrapment efficiency and particle size are shown in Figure 4 and Figure 5, respectively.

#### 3.4.3. Effect of Addition Rate

Addition rates showed a direct relationship with drug entrapment efficiency and particle sizes of the niosomes. The data demonstrated that by increasing the addition rate from 0.25 to 1 mL/min, higher drug entrapment efficiency as well as lower particle sizes were achieved. The effect of addition rate on entrapment efficiency and particle size of niosomes is shown in Figure 6. 

### 3.5. Entrapment Efficiency (%)

A higher drug entrapment efficiency will be able to give better niosomal formulation by providing higher drug concentrations delivered to the target site in the patient. Vesicular entrapment efficiency is a crucial parameter that conveys stability of the vesicles and depends upon the multiple factors and combinations that were used to manufacture the niosomal dispersion. The entrapment efficiencies of the various formulations used in this study are provided in Table 3. From the data, it is clear that entrapment efficiency depends on multiple variables. The entrapment efficiency of niosome formulations ranged from 63.90% to 95.58% in all the formulations. 

As shown in Table 3, the entrapment efficiency of niosomes formed from diethyl ether:methanol (75:25) was found to be high compared with niosomes prepared by the use of any other solvent systems. Changes in drug concentrations had no impact on entrapment efficiency. The formulations containing 2:4:2, 4:4:2, and 2:4:3 ratios of drug:surfactant:cholesterol prepared with diethyl ether:methanol showed high entrapment efficiency in comparison with the formulations containing 3:4:2, 2:2:2, 2:3:2, 2:5:2, 2:6:2, 2:7:2, 2:4:4 and 2:4:5 ratios of drug:surfactant:cholesterol. External phase temperature, external phase volume and internal phase volume did not conclusively impact theentrapment efficiency of niosomes. For mixing time and speed, entrapment efficiency was directly correlated and increased with an increase in mixing time and speed. The organic phase addition rate also followed identical trends as mixing properties and showed increasing entrapment efficiencies with increasing addition rates. 

### 3.6. Vesicle Sizes and Polydispersity Index

Particle sizes and polydispersity indices of niosomes formulations ranged from 154.40 to 919.87 nm and 0.144 to 0.441, respectively. As shown in Table 3, the particle size of DND-3 used an organic phase containing diethyl ether:methanol (75:25) and was found to be an ideal niosome that could be used for topical skin applications. When adding excess amounts of both cholesterol and desoximetasone, a marked niosome particle size increase was observed. This behavior may be explained by further understanding the interactions that both the drug and cholesterol have on the niosome vesicle bilayer [40,43,44]. The formulation showed an increasing trend of particle sizes with increasing cholesterol content. This behavior may be explained by the fact that higher cholesterol content imparts rigidity to the niosomal matrix and accumulates in the bilayer, which increases the hydrodynamic diameter and particle size of niosomes [46]. On a regular basis, the cholesterol mechanism of action is twofold: (a) cholesterol increases the chain order of liquid state bilayers and (b) cholesterol reduces the chain order of gel state bilayers. As cholesterol concentration increases, the gel state is gradually transformed to a liquid ordered phase. An increase in cholesterol content of the niosomes bilayers results in a decrease in the release rate of entrapped drug and therefore an increase in the bilayer rigidity [47]. External phase temperature, external phase volume and internal phase volume did not have any major role in determining particle size. Both mixing time and speed had a direct impact on particle size. A decreasing trend was observed for particle sizes with increasing mixing time and speed. The organic phase addition rate also followed a similar behavior to the mixing parameters.

## 4. Conclusions

The experimental design conditions demonstrated that selecting the appropriate surfactant and cholesterol leads to the formation of desoximetasone-loaded niosomes with desired particle sizes and entrapment efficiencies for topical use. The critical material attributes (surfactant, cholesterol amounts) and critical process parameters (mixing time and speed) were identified and used to understand the critical quality attributes of topical niosome formulations. Mixing time and speed increases during the niosome formulation preparation resulted in a higher entrapment efficiency and lower particle sizes. Desoximetasone-loaded niosomes with desired particle size ranges and entrapment efficiencies for topical administration routes may be obtained by carefully selecting the correct combination of components. Based on CMAs and CPPs, a total of 11 variables were considered for the study. A summary of fixed and impacting parameters is given in Table 4.

This work establishes the fundamental foundations which may be explored in order to develop a lean and robust manufacturing process by optimizing QbD elements. This extensive study provides a comprehensive understanding of drug product design. Future studies are needed with the assistance of advanced scientific design tools in order to promote more precise control of formulation outcomes.

## Figures and Tables

**Figure 1 pharmaceutics-12-00246-f001:**
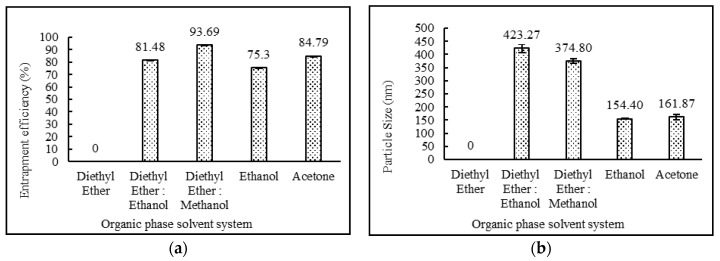
Effect of organic phase system on (**a**) entrapment efficiency (%) and (**b**) particle size (nm) of niosomes. (*N* = 3, mean ± SD).

**Figure 2 pharmaceutics-12-00246-f002:**
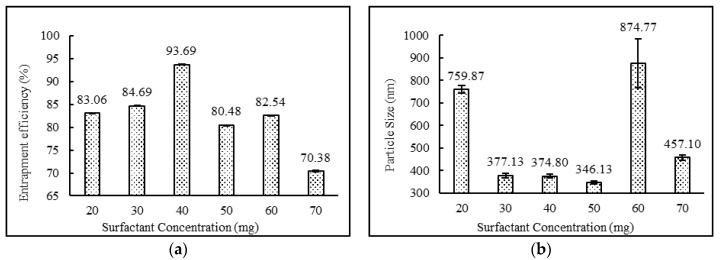
Effect of surfactant concentration (mg) on (**a**) entrapment efficiency (%) and (**b**) particle size (nm) of niosomes. (N = 3, mean ± SD).

**Figure 3 pharmaceutics-12-00246-f003:**
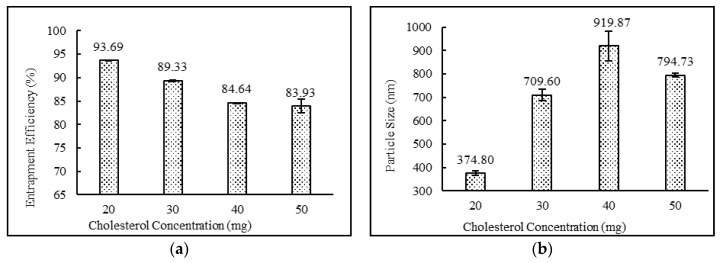
Effect of cholesterol concentration (mg) on (**a**) entrapment efficiency (%) and (**b**) particle size (nm) of niosomes. (*N* = 3, mean ± SD)

**Figure 4 pharmaceutics-12-00246-f004:**
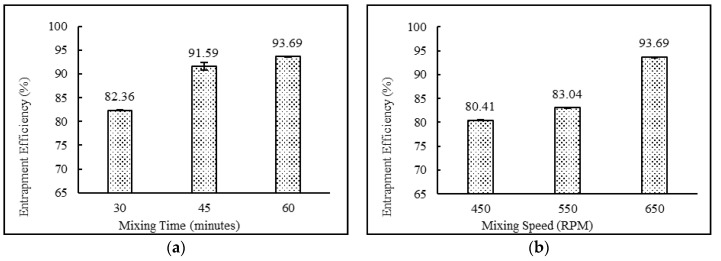
Effect of (**a**) mixing time (minutes) and (**b**) mixing speed (rpm) on entrapment efficiency (%) of niosomes. **(***N* = 3, mean ± SD).

**Figure 5 pharmaceutics-12-00246-f005:**
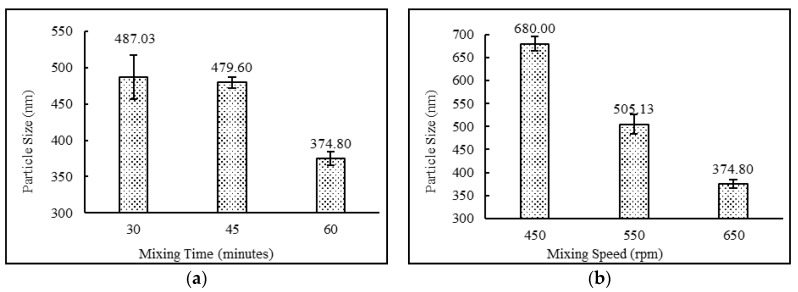
Effect of (**a**) mixing time (minutes) and (**b**) mixing speed (rpm) on particle size (nm) of niosomes. (*N* = 3, mean ± SD).

**Figure 6 pharmaceutics-12-00246-f006:**
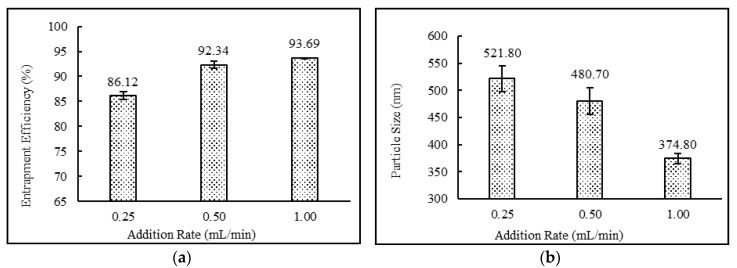
Effect of addition rate (ml/min) on (**a**) entrapment efficiency (%) and (**b**) particle size (nm) of niosomes. (*N* = 3, mean ± SD).

**Table 1 pharmaceutics-12-00246-t001:** Critical material attributes (CMAs) and critical process parameters (CPPs) for the formulation design.

No	Variables: CMAs and CPPs	Parameters
1	Organic phase preparation (Solvent)	a. Diethyl Ether
		b. Diethyl Ether: Ethanol (80:20)
		c. Diethyl Ether: Methanol (75:25)
		d. Ethanol
		e. Acetone
2	Drug concentration (mg)	a. 20 mg (0.2%)
		b. 30 mg (0.3%)
		c. 40 mg (0.4%)
3	Surfactant concentration (mg)	a. 20 mg (0.2%))
		b. 30 mg (0.3%)
		c. 40 mg (0.4%
		d. 50 mg (0.5%)
		e. 60 mg (0.6%)
		f. 70 mg (0.7%)
4	Cholesterol concentration (mg)	a. 20 mg (0.2%)
		b. 30 mg (0.3%)
		c. 40 mg (0.4%)
		d. 50 mg (0.5%)
5	Selection of lipid	a. Stearic acid
		b. Stearylamines
		c. No lipids
6	External phase temperature (°C)	a. 55 °C
		b. 65 °C
		c. 75 °C
7	External phase volume (mL)	a. 10 mL
		b. 20 mL
		c. 30 mL
8	Internal phase volume (mL)	a. 10 mL
		b. 15 mL
		c. 20 mL
9	Mixing speed (rpm)	a. 450 rpm
		b. 550 rpm
		c. 650 rpm
10	Mixing time (minutes)	a. 30 min
		b. 45 min
		c. 60 min
11	Addition Rate (mL/min)	a. 0.25 mL/min
		b. 0.50 mL/min
		c. 1.00 mL/min

**Table 2 pharmaceutics-12-00246-t002:** Quality by Design (QbD)—Niosome Batch Experimental Design Table.

Batch ID #	Critical Material Attributes					Critical Processing Parameters					
	Drug (mg)	Organic Phase Composition	Span 60 (mg)	Cholesterol (mg)	Stearic Acid (mg)	External Phase Temperature (°C)	External Phase Volume (mL)	Internal Phase Volume (mL)	Mixing Speed (RPM)	Mixing Time (Minutes)	Add. Rate (mL/Min)
DND-1	20	**diethyl ether**	40	20	5	65	20	10	650	60	1.00
DND-2	20	**diethyl ether : ethanol**	40	20	5	65	20	10	650	60	1.00
DND-3	20	**diethyl ether : methanol**	40	20	5	65	20	10	650	60	1.00
DND-4	20	**ethanol**	40	20	5	65	20	10	650	60	1.00
DND-5	20	**acetone**	40	20	5	65	20	10	650	60	1.00
DND-6	**30**	diethyl ether : methanol	40	20	5	65	20	10	650	60	1.00
DND-7	**40**	diethyl ether : methanol	40	20	5	65	20	10	650	60	1.00
DND-8	20	diethyl ether : methanol	**20**	20	5	65	20	10	650	60	1.00
DND-9	20	diethyl ether : methanol	**30**	20	5	65	20	10	650	60	1.00
DND-10	20	diethyl ether : methanol	**50**	20	5	65	20	10	650	60	1.00
DND-11	20	diethyl ether : methanol	**60**	20	5	65	20	10	650	60	1.00
DND-12	20	diethyl ether : methanol	**70**	20	5	65	20	10	650	60	1.00
DND-13	20	diethyl ether : methanol	40	**30**	5	65	20	10	650	60	1.00
DND-14	20	diethyl ether : methanol	40	**40**	5	65	20	10	650	60	1.00
DND-15	20	diethyl ether : methanol	40	**50**	5	65	20	10	650	60	1.00
DND-16	20	diethyl ether : methanol	40	20	**5 (SA)**	65	20	10	650	60	1.00
DND-17	20	diethyl ether : methanol	40	20	**N/A**	65	20	10	650	60	1.00
DND-18	20	diethyl ether : methanol	40	20	5	**55**	20	10	650	60	1.00
DND-19	20	diethyl ether : methanol	40	20	5	**75**	20	10	650	60	1.00
DND-20	20	diethyl ether : methanol	40	20	5	65	**10**	10	650	60	1.00
DND-21	20	diethyl ether : methanol	40	20	5	65	**30**	10	650	60	1.00
DND-22	20	diethyl ether : methanol	40	20	5	65	20	**15**	650	60	1.00
DND-23	20	diethyl ether : methanol	40	20	5	65	20	**20**	650	60	1.00
DND-24	20	diethyl ether : methanol	40	20	5	65	20	10	**450**	60	1.00
DND-25	20	diethyl ether : methanol	40	20	5	65	20	10	**550**	60	1.00
DND-26	20	diethyl ether : methanol	40	20	5	65	20	10	650	**30**	1.00
DND-27	20	diethyl ether : methanol	40	20	5	65	20	10	650	**45**	1.00
DND-28	20	diethyl ether : methanol	40	20	5	65	20	10	650	60	**0.25**
DND-29	20	diethyl ether : methanol	40	20	5	65	20	10	650	60	**0.50**

**Table 3 pharmaceutics-12-00246-t003:** Summary of Results for QbD—Niosomal Dispersions.

Batch ID #	RESULTS (NIOSOMAL DISPERSION)					
	Dispersion Formed or Not	Organoleptic Properties	Entrapment Efficiency (%)	Particle Size (nm)	Polydispersity Index	Zeta Potential (mV)
DND-1	No	N/A	N/A	N/A	N/A	N/A
DND-2	Yes	White Milky	81.48 ± 0.07	423.27 ± 16.48	0.294 ± 0.04	−75.63 ± 0.61
DND-3	Yes	White Milky	93.69 ± 0.05	374.80 ± 9.48	0.289 ± 0.01	−63.83 ± 4.26
DND-4	Yes	White Milky	75.30 ± 0.06	154.40 ± 1.47	0.144 ± 0.02	−54.13 ± 1.16
DND-5	Yes	White Milky	84.79 ± 0.06	161.87 ± 10.83	0.330 ± 0.03	−52.10 ± 1.51
DND-6	Yes	White Milky	89.38 ± 0.03	411.20 ± 22.53	0.235 ± 0.03	−43.03 ± 0.59
DND-7	Yes	White Milky	91.43 ± 0.01	655.07 ± 46.64	0.276 ± 0.02	−46.30 ± 0.87
DND-8	Yes	White Milky	83.06 ± 0.04	759.87 ± 16.66	0.316 ± 0.01	−51.03 ± 0.15
DND-9	Yes	White Milky	84.69 ± 0.03	377.13 ± 10.90	0.322 ± 0.02	−69.53 ± 0.40
DND-10	Yes	White Milky	80.48 ± 0.03	346.13 ± 6.03	0.303 ± 0.01	−58.30 ± 0.66
DND-11	Yes	White Milky	82.54 ± 0.05	874.77 ± 109.42	0.394 ± 0.01	−57.53 ± 1.86
DND-12	Yes	White Milky	70.38 ± 0.20	457.10 ± 12.21	0.328 ± 0.03	−38.50 ± 2.14
DND-13	Yes	White Milky	89.33 ± 0.11	709.60 ± 24.44	0.345 ± 0.01	−81.67 ± 2.30
DND-14	Yes	White Milky	84.64 ± 0.03	919.87 ± 64.56	0.371 ± 0.01	−54.67 ± 1.20
DND-15	Yes	White Milky	83.93 ± 1.42	794.73 ± 9.07	0.324 ± 0.01	−57.60 ± 2.01
DND-16	Yes	White Milky	82.71 ± 0.11	56237.83 ± N/A	N/A	11.43 ± 0.76
DND-17	Yes	White Milky	63.90 ± 0.12	616.33 ± 7.02	0.348 ± 0.01	−41.53 ± 0.72
DND-18	Yes	White Milky	80.75 ± 0.02	435.77 ± 27.99	0.441 ± 0.05	−40.23 ± 1.30
DND-19	Yes	White Milky	70.41 ± 0.05	460.95 ± 12.07	0.320 ± 0.02	−60.07 ± 0.78
DND-20	Yes	White Milky	95.58 ± 0.03	355.30 ± 8.11	0.282 ± 0.01	−58.53 ± 1.33
DND-21	Yes	White Milky	74.56 ± 0.06	823.77 ± 19.59	0.334 ± 0.01	−53.77 ± 0.95
DND-22	Yes	White Milky	77.68 ± 0.06	444.37 ± 0.11	0.318 ± 0.01	−51.17 ± 0.81
DND-23	Yes	White Milky	76.48 ± 0.10	439.70 ± 56.28	0.273 ± 0.03	−47.23 ± 1.75
DND-24	Yes	White Milky	80.41 ± 0.09	680.00 ± 15.57	0.292 ± 0.01	−34.13 ± 1.17
DND-25	Yes	White Milky	83.04 ± 0.06	505.13 ± 20.78	0.261 ± 0.03	−48.73 ± 1.10
DND-26	Yes	White Milky	82.36 ± 0.03	487.03 ± 30.69	0.304 ± 0.01	−48.73 ± 0.81
DND-27	Yes	White Milky	91.59 ± 0.84	479.60 ± 7.79	0.310 ± 0.02	−49.53 ± 2.48
DND-28	Yes	White Milky	86.12 ± 0.78	521.80 ± 23.80	0.310 ± 0.00	−54.17 ± 1.25
DND-29	Yes	White Milky	92.34 ± 0.68	480.70 ± 24.61	0.302 ± 0.03	−58.60 ± 0.44

**Table 4 pharmaceutics-12-00246-t004:** Summary of fixed and impacting parameters for desoximetasone niosomes.

Fixed Parameters	Impacting Parameters
Method of preparation (Diethyl ether : methanol :: 75 : 25)	Surfactant concentration
Drug concentration (20 mg)	Cholesterol concentration
Lipid (Stearic Acid—5 mg)	Mixing speed
External phase temperature (65 °C)	Mixing time
External phase volume (20 mL)	Addition rate
Internal phase volume (10 mL)

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
