# Peer review of "Evaluations of Quality by Design (QbD) Elements Impact for Developing Niosomes as a Promising Topical Drug Delivery Platform"

_pharmaceutics, 2020, doi:10.3390/pharmaceutics12030246_

Round 1
Reviewer 1 Report
The aim of this work was to evaluate the criteria that impact the formulation of niosomes in terms of vesicle size and encapsulation efficiency of desoximetasone. Various experimental parameters were investigated and an « ideal » combination was found. The influence of the different parameters was also explained to support the results. For these reasons, the objectives of the work were reached. However, some details should be given by the authors to complete this study.
Abstract
In the abstract, several abbreviations are not explained (CMA, CPP and QTPP) and make the understanding difficult. Moreover, the last sentence (line 31-32) assumes that all the parameters of niosomes formulation were optimized for topical application for all kinds of API ; the authors should specify that it concerns only desoximetasone.
Introduction
The introduction covers a lot of different interesting points. However, as the main aim of the work is to study the quality by design process for niosomes, more focus should be put on the importance of this approach in the topical delivery. In other words, the authors should explain what are the advantages and consequences of this kind of approach.
The authors make different assumptions about the role of nanocarriers in topical drug delivery and considered that they enable « localized amount of drug to a targeted site » (line 60) and create « a local depot in deeper layer of skin ». This supposes that the niosomes will stay intact and penetrate the skin, which is really unlikely. Considering the structure and the composition of the skin and the size of nanocarriers, it is more plausible that they accumulate in the surface of skin. Therefore, the authors should revise this part.
Materials and methods
The authors should state whether the HPLC method was validated with the ICH guideline. The same remark can be applied for the method to characterize the « drug entrapment efficiency » (line 198). Did the authors validated the speed and the time of the centrifugation to ensure the total sedimentation of niosomes?
Results and discussion
The authors mentioned that the measurements of the encapsulation efficiency and particle size were done in triplicate. Does it mean that the measurement was done in triplicate or that three formulations were prepared ?
As in the introduction, the authors make a really controversial assumption by explaining that « small particle may not stay in skin layers and may penetrate into the systemic circulation before releasing the drug at site of application » (line 248-251). The authors should be more precise and aware about the role of nanocarriers in dermal delivery, because they are claiming that nanocarriers could permeate in the systemic circulation and this is not necessarily the case.
In general, the same terminology as in Table 2 for the name of the formulations (batch ID number) should be used all along the text to facilitate the follow up of the work.
In the discussion about the « effect of drug, surfactant and cholesterol concentration », the discussion and the results presented in Figure 2 do not correspond. In fact, in line 263-263, the authors explain that an increase in surfactant concentration increases the encapsulation efficiency. However, it is not the case in the Figure 2 as at 50 mg of surfactant, the encapsulation efficiency descreases. The authors should find another way to represent these results.
Moreover, in this part, the authors say that « higher cholesterol increases the bilayer hydrophobicity and stability and which negatively impact the ability of the hydrophobic drug to get trapped into vesicle bilayer as will reduce the penetration » (line 265-268). The authors should give additional information about why and how an increase in the bilayer hydrophobicity decreases the entrapment of hydrophobic drug.
In table 3, the authors should add the SD for all the values.
Finally, a table encompassing all the ideal conditions for the niosomes formulation of desoximethasone should be added as a summary.
Author Response
Please see attached file for the detail review.

Reviewer 2 Report
The manuscript describes an evaluation of the parameters that influence the preparation of niosomes.
The aim of the work is good and the study is correctly designed. However, in my opinion the manuscript is lacking in innovation of research. In fact, it describes, in a systematic way, the preformulative studies that are normally carried out to optimize one or more formulations.
In my opinion the new information contained in the manuscript may at most be the subject of a "short communication”.
The manuscript is generally well written and adequately referenced, however some sentences could be clarified or referenced.
In particular:
The introduction, although well written, is very generic and it is a description of the history and evolution of "drug delivery". It seems an introductory lesson for students. The only part that could be appropriate for a scientific manuscript is that concerning the last 20-25 lines. This is also evidenced by many references that are old.
In Materials & Methods:
the brand and type of column used in the HPLC analysis is missing
In Results and Discussion:
The affirmation of the Authors in the sentence on p. 8 lines 248-251 must be supported by the literature, otherwise it is just a supposition without scientific basis.
The sentence on pg. 12 lines 358-359 is ambiguous. The niosomes obtained are optimal for topical skin applications, not the solvent mixture.
Pag. 12 lines from 366 to 370: the sentences are unclear.
Reference 40 must be corrected
Reference 43 it is a very old edition of the Remington's pharmaceutical sciences
Author Response
Please see the attached file for the detail response.

Reviewer 3 Report
The manuscript "Evaluations of Quality by Design (QbD) Elements 2 Impact for Developing Niosomes as a Promising 3 Topical Drug Delivery Platform" is an interesting article which deals with the development of niosomes prepared with surfactants and cholesterol and tailored to deliver desoximetasone via topical application. The manuscript provides the fundamental foundations which may be explored in order to develop a lean and robust manufacturing process by optimizing Quality and Design elements. This The topic of the articles is interesting, it is well written, and the abstract underline the promising challenges of the study. I recommend publication in the Pharmaceutics after minor revisions.
In the abstract, please report the significate of CMA, CPP, QTPP, QbD, PDI.
Additionally, an excessive amount of abbreviations is reported (CMA, CPP, QTPP, QbD, PDI, API, DND, DoE, CMA) in the whole manuscript, and they make very difficult the text reading and compression.
In the Introduction, lines 88-89, the sentence “The addition of cholesterol provides the rigidity and orientation of the niosomal bilayer resulting in less leaky niosomes” is limitative because the cholesterol can modify the bilayer assembly increasing or reducing the fluidity (see Sung-Tae Yang et al., The Role of Cholesterol in Membrane Fusion).
In the Introduction, line 101, the ability of niosomes to improve cellular up-take must be added and discuss (see Manconi et., Development of novel diolein-niosomes for cutaneous delivery of tretinoin: Influence of formulation and in vitro assessment).
In the experimental section, line 174, Table 1, the authors reported drug, surfactant and cholesterol concentration but only their amount (mg) has been reported. The actual concentrations of drug, surfactant and cholesterol must be reported.
Author Response

(The authors gave the same response as above.)

Round 2
Reviewer 1 Report
The revised version of the manuscript has been improved and we are satisfied by the majority of the responses given by the authors.
However, the authors have not addressed the points raised with respect to niosome penetration in the stratum corneum and the effect of niosome size on drug delivery. The authors seem to suggest that niosomes can penetrate across the skin and even reach the systemic circulation. This theory is far from being accepted in the topical drug delivery field. Regarding the structure of the stratum corneum, and given the composition and size of the niosomes, why do the authors consider that intact niosomes can cross this layer?
Niosomes, like other nanocarriers, might be able to penetrate deeper via the follicular pathway. However, the diameter of a hair follicle is on the micrometer scale. Thus, all nanocarriers could potentially penetrate by this pathway (different studies were performed to understand which are the optimal parameters for a nanocarrier to reach this pathway). The authors should be aware and amke clear that the internalization and the targeted behavior of nanocarriers during cell-based assay is different from skin delivery studies.
Author Response
We would like to thank the reviewers for taking time out of their busy schedule to review our manuscript. We greatly appreciate your comments and have made changes to the manuscript in order to address your concerns. We hope that you find them satisfactory.
-----------------------------------------------------------------------------------------------------------------
Reviewer# 1:
Response: Thank you to this reviewer for your feedback. Upon recommendation we have emend this part of the manuscript and re-written the sentence. Please see page 8 lines 265-266.
Reviewer 2 Report
The revised version of the manuscript has been improved and I am satisfied by the corrections given by the authors regarding the clarification of some sentences.
However, the authors affirm that reporting a background of niosomes at the beginning of the introduction is necessary. This would be true if they did not start from 1970 and treated the topic in a generic way. A treatment of the state of the art of the delivery of dexamethasone to the skin through nanocarriers would be much more appropriate.
By way of example, some recent publications are reported:
Nazari-Vanani, et al. (2019), Nanomedicine, and Biotechnology, 47:1, 420-426, DOI: 10.1080/21691401.2018.1559179
Arafa et al. Sci Rep 8, 18056 (2018). https://doi.org/10.1038/s41598-018-37157-7
Sahu et al. Soft Matter. 2020 Jan 23. doi: 10.1039/c9sm02416f.
The authors perhaps forget that the manuscript and, in general, the journal are addressed to a reader who already has much experience in the field.
The observation relating to the old references refers precisely to those that support the introduction part concerning "the history" of nanocarriers, namely:
[4] Allen, T.M., Liposomal drug formulations. Drugs, 1998. 56(5): p. 747-756.
[9] Handjani-Vila, R.M., et al., Dispersions of lamellar phases of non-ionic lipids in cosmetic products. Int J Cosmet Sci, 1979. 1(5): p. 303-14
[20] Baillie, A., et al., Non-ionic surfactant vesicles, niosomes, as a delivery system for the anti-leishmanial drug, sodium stibogluconate. Journal of pharmacy and pharmacology, 1986. 38(7): p. 502-505.
[27] Uchegbu, I.F. and S.P. Vyas, Non-ionic surfactant based vesicles (niosomes) in drug delivery. International journal of pharmaceutics, 1998. 172(1-2): p. 33-70.
[55] BL, S., The Physical Chemistry of Membranes. New York: Alan and Unwin and Soloman Press, 1985. Silver p. 209-30.
In addition, The latter must be corrected
Finally, the whole paragraph "References" must be revised as it does not respect the requests of the journal. Indeed, all author names in a reference citation should be included.
Author Response
We would like to thank the reviewers for taking time out of their busy schedule to review our manuscript. We greatly appreciate your comments and have made changes to the manuscript in order to address your concerns. We hope that you find them satisfactory.
-----------------------------------------------------------------------------------------------------------------
Reviewer# 2:
Response: Thank you to this reviewer for your feedback. Upon recommendation we have revised part of the manuscript and replaced old references. Please see page 3 lines 120-125.
We have update all the reference as per MDPI journal requirement. Please see page 14-17 lines 417-558.
Round 3
Reviewer 2 Report
The authors accepted the suggestions relating to updating and correcting the references, however they did not accept the suggestion on remodeling the introduction towards a reduction of the part on "history of niosomes", already suggested in the first revision.
Less than an additional sentence concerning the topical application of desoximetasone, the introduction has remained identical to the previous version.
Author Response
We would like to thank the reviewers for taking time out of their busy schedule to review our manuscript. We greatly appreciate your comments and have made changes to the manuscript in order to address your concerns. We hope that you find them satisfactory.
-----------------------------------------------------------------------------------------------------------------
Reviewer# 2:
Response: Thank you for mentioning this point. Upon recommendation we have re-written the part of introduction and have reduced the part on “history of niosomes”. Please see page 2 lines 68-76.